# The Silent Threat of Human Metapneumovirus: Clinical Challenges and Diagnostic Insights from a Severe Pneumonia Case

**DOI:** 10.3390/microorganisms13010073

**Published:** 2025-01-02

**Authors:** Rubens Carmo Costa-Filho, Felipe Saddy, João Luiz Ferreira Costa, Leandro Reis Tavares, Hugo Caire Castro Faria Neto

**Affiliations:** 1Immunopharmacology Laboratory, Oswaldo Cruz Institute/FIOCRUZ, Rio de Janeiro 21040-361, RJ, Brazil; hugocfneto@gmail.com; 2Intensive Care Unit, Pró-Cardíaco Hospital, Rio de Janeiro 21040-361, RJ, Brazil; fsaddy@gmail.com (F.S.); joaoluizferreiracosta@gmail.com (J.L.F.C.); 3D’Or Institute Research and Education (IDOR), Rio de Janeiro 21040-361, RJ, Brazil; lreistavares@gmail.com

**Keywords:** human metapneumovirus (hMPV), community-acquired pneumonia (CAP), rapid molecular diagnostics, immunocompetent adults, emerging respiratory viruses

## Abstract

Background: Human metapneumovirus (hMPV) is a respiratory pathogen that has gained increasing recognition due to advancements in molecular diagnostic tools, which have improved its detection and characterization. While severe disease manifestations are traditionally associated with pediatric, elderly, or immunocompromised patients, hMPV-related pneumonia in immunocompetent adults remains underexplored. Methods: This case report describes a 68-year-old male who developed severe community-acquired pneumonia (CAP) caused by hMPV despite being immunocompetent and having no significant comorbidities. The diagnosis was confirmed via multiplex RT-PCR, excluding bacterial and viral coinfections. Laboratory and imaging findings supported a viral etiology, while empirical antibiotics were initially prescribed due to diagnostic uncertainty. Results: The patient presented with respiratory symptoms that progressed to hypoxia, productive cough, and fatigue, requiring hospitalization. Imaging revealed bilateral ground-glass opacities and consolidations typical of viral pneumonia. Molecular diagnostics enabled accurate pathogen identification and guiding appropriate management, and the patient fully recovered with supportive care. Conclusion: This case underscores the importance of rapid molecular diagnostics for hMPV, reducing unnecessary antibiotics and enhancing management. Routine incorporation of hMPV testing into clinical protocols could improve the diagnosis and resource use. The development of vaccines, such as the IVX-A12 in phase II trials, offers hope for targeted prevention, underscoring the need for further research and therapeutic innovations.

## 1. Introduction

Human metapneumovirus (hMPV) is an increasingly recognized respiratory pathogen with significant clinical and public health implications. First identified in 2001, serological evidence has indicated its presence in human populations since the 1950s [1]. The family Pneumoviridae comprises two genera: *Metapneumovirus*, which includes Human metapneumovirus (hMPV), and *Orthopneumovirus*, which includes Respiratory syncytial virus (RSV). This classification underscores the close relationship between these viruses while highlighting their distinct clinical and epidemiological features. Clinical and genetic similarities complicate the diagnosis of hMPV, contributing to its under-recognition in both research and clinical practice, where routine testing remains limited. hMPV is associated with a spectrum of respiratory illnesses ranging from mild upper respiratory tract infections, such as the common cold, to severe lower respiratory tract conditions, including bronchitis, pneumonia, and asthma exacerbations. Severe disease manifestations are more frequently observed in young children, elderly individuals, and immunocompromised patients [2]. However, severe hMPV infections have been extensively documented not only in elderly patients with comorbidities but also in healthy adults across all age groups [3,4]. These findings underscore the need for increased clinical awareness of hMPV in diverse patient populations. 

Studies characterizing severe hMPV-related illnesses have predominantly focused on pediatric populations, leaving data on adult cases that are infrequent and fragmented. For example, one study reported that 12% of hospitalized patients with hMPV required ICU admission, with 11% requiring ventilator support [1]. Retrospective cohort studies and case reports have predominantly focused on children and immunocompromised populations. The 2024 PERCH study [5] (Pneumonia Etiology Research for Child Health), conducted across high-burden settings in Africa and Asia, identified hMPV as the second leading cause of severe pneumonia in children under five, following RSV. The study highlighted hMPV’s frequent co-occurrence with bacterial pathogens and its significant contribution to pneumonia-related morbidity and mortality.

Similarly, Kuang et al. [6] analyzed over 155,000 pediatric hospitalizations for acute respiratory tract infections (ARTI) over 12 years, showing that severe hMPV cases were most prevalent in infants under one year of age and in children with comorbidities, with ICU admissions in 2.34% of hMPV-positive pediatric cases. This emphasis on pediatric and immunosuppressed populations has left significant gaps in the understanding of hMPV’s clinical impact in immunocompetent adults despite evidence suggesting that such cases, while less frequent, may lead to severe outcomes. This case report raises awareness of human metapneumovirus (hMPV) as a potential cause of severe pneumonia in immunocompetent older adults, highlighting the need for greater clinical recognition and consideration of this pathogen in differential diagnoses.

Consequently, the prevailing assumption among healthcare providers is that only immunocompromised adults or those with significant comorbidities are vulnerable to severe outcomes. However, emerging evidence challenges this perception. Walsh et al. [7] conducted a four-year prospective study of approximately 1400 hospitalized adults. They found that hMPV accounted for 8% of acute respiratory illnesses requiring hospitalization—a rate comparable to that of RSV and higher than that of influenza A in the same cohort. The study highlighted significant morbidity associated with hMPV, including an average hospital stay of nine days, ICU admission in 13.2% of cases, and a mortality rate slightly lower than those of RSV and influenza A. These findings underscore the potential of hMPV to cause severe disease, even in immunocompetent adults.

Here, we present a case of severe hMPV-related pneumonia in an immunocompetent adult, contributing to the growing body of evidence that hMPV can impose a significant clinical burden on populations traditionally considered at low risk.

## 2. Case Presentation

A 68-year-old male with a history of mild systemic arterial hypertension, controlled with candesartan, dyslipidemia managed with simvastatin, and chronic aspirin use, presented with progressively worsening respiratory symptoms. The patient was physically active, running three times per week, non-obese, a non-smoker, and had no history of diabetes or other significant comorbidities.

The symptoms began with mild odynophagia and intermittent dry cough (D1). Over the next few days, the symptoms progressed to rhinorrhea, nasal congestion, allodynia, nocturnal sweating (D2), and eventually hyposmia and hypogeusia. A self-administered SARS-CoV-2 nasopharyngeal swab test on (D2) returned negative results. On (D3), empirical therapy with amoxicillin-clavulanate, clarithromycin, and oral prednisone was initiated (D4); however, the patient’s condition worsened. By D5, he experienced intense coughing and bronchospasm, followed by a productive cough with thick, yellow-brown sputum, along with myalgia, headache, and fatigue.

On (D7), he was admitted to the Emergency Department with SpO_2_ levels of 91–92% on room air, tachypnea, fatigue, an axillary temperature of 37.8 °C, and blood pressure of 140/80 mmHg. (Figure 1). Laboratory findings included elevated inflammatory markers (C-reactive protein, 7.6 mg/dL; D-dimer, 870 ng/mL), mild transaminase elevation, and thrombocytopenia, while leukocyte counts and procalcitonin levels remained normal, suggesting a viral etiology. Imaging studies revealed significant pulmonary involvement (Figure 2).

Axial computed tomography (CT) of the thorax revealed mild bilateral pleural effusions, small centrilobular opacities with multilobar distribution, and ground-glass opacities, some of which were coalescent and interspersed with areas of consolidation. These findings predominantly affected the left upper and lower lobes, involving the pulmonary interstitium and peripheral airways (acinus and bronchioles), which is consistent with a viral pneumonia pattern.

Additionally, CT scans of the paranasal sinuses showed mucosal thickening associated with secretions in the frontal sinuses and ethmoidal cells bilaterally, along with mild mucosal thickening in the maxillary sinuses, consistent with sinusitis (Figure 3). This presentation strongly suggests an underlying infection or an inflammatory process.

During hospitalization, the patient received intravenous moxifloxacin, hydration, nebulization therapy with oxygen, ipratropium bromide, salbutamol, and physiotherapy to relieve the bronchospasm and improve expectoration. He stabilized within 24 h and was discharged on oral moxifloxacin, with instructions for outpatient management focused on symptomatic care. Over the next week, he experienced gradual improvement, with resolution of bronchospasm and cough, achieving a slow recovery by Day 14 post-discharge.

Further diagnostic evaluations in the emergency department included molecular testing via a multiplex real-time polymerase chain reaction (RT-PCR) panel (Table 1), which identified human metapneumovirus (hMPV) as the sole pathogen. Coinfections with bacterial, atypical, or other viral agents were excluded, as were qualitative SARS-CoV-2 findings obtained using an immunochromatographic assay (Abbott). This molecular methodology allowed for rapid pathogen identification, with results obtained in approximately one hour, guiding the clinical approach and confirming the viral etiology of the disease. Procalcitonin levels were within normal limits (0.054 ng/mL), supporting a viral rather than a bacterial etiology. Notable findings included elevated inflammatory markers, such as C-reactive protein (CRP) (7.6 mg/dL) and D-dimer (870 ng/mL), mild transaminase elevation (AST 45 U/L; ALT 39 U/L), and renal function parameters mildly above the reference range (creatinine 1.47 mg/dL; urea 41 mg/dL). Venous blood gas analysis was performed instead of arterial blood gas analysis due to patient-specific considerations, including clinical stability, the less invasive nature of the procedure, and the patient’s refusal of arterial sampling. The analysis revealed metabolic compensation, with bicarbonate levels of 30.4 mmol/L and a venous pH of 7.41. Hematological findings showed mild thrombocytopenia (134,000/mm^3^) and normal leukocyte counts, with a neutrophilic predominance (69%). Platelet levels remained stable on the subsequent day, with minimal fluctuations in inflammatory and hematological markers.

This case highlights the potential severity of hMPV infection even in adults without significant comorbidities. Contentin et al. [8] reported similar cases of acute respiratory distress syndrome (ARDS) secondary to hMPV, reinforcing the pathogen’s ability to provoke severe complications in immunocompetent individuals.

The multiplex RT-PCR assay identified human metapneumovirus (hMPV) as the sole pathogen. No bacterial or viral coinfections were detected. The methodology amplifies specific DNA/RNA sequences using fluorescent probes, providing rapid results and confirming the viral etiology. These findings highlight the utility of molecular diagnostics in guiding targeted clinical management.

## 3. Discussion

### 3.1. Community-Acquired Pneumonia and hMPV

Community-acquired pneumonia (CAP) remains a significant global cause of morbidity and mortality, disproportionately affecting children under 5 years, adults over 65 years, and individuals with chronic comorbidities [9]. Vaccination programs targeting bacterial pathogens, such as *Streptococcus pneumoniae* and *Haemophilus influenzae type b*, have significantly reduced bacterial CAP; however, knowledge gaps persist regarding the microbial etiology of severe CAP requiring hospitalization. This case highlights the growing recognition of human metapneumovirus (hMPV) as a frequent cause of CAP across different age groups, particularly as bacterial infections decline [10]. Additionally, the patient adhered to comprehensive vaccination protocols, including pneumococcal, RSV, influenza, and recent mRNA SARS-CoV-2 vaccines.

### 3.2. Severity of hMPV in Immunocompetent Adults

hMPV has traditionally been associated with severe respiratory infections in pediatric, elderly, and immunocompromised populations [7,11]. However, this case demonstrates that hMPV can cause significant pneumonia in immunocompetent adults without comorbidities. Walsh et al. [7] reported that hMPV contributes to 8% of adult hospitalizations for acute respiratory illnesses, with rates comparable to RSV and higher than influenza A. Our patient presented with severe pneumonia, yet without bacterial markers such as leukocytosis with a left shift or elevated procalcitonin, underscoring a viral etiology. Despite molecular diagnostics confirming hMPV as the sole pathogen, empirical antibiotic therapy with intravenous moxifloxacin was initiated for two days. After discharge, oral administration was continued for over 10 days, reflecting the persistent diagnostic uncertainty in distinguishing viral from bacterial infections [12,13].

#### 3.2.1. Role of Rapid Molecular Diagnostics

Rapid molecular diagnostics, specifically the FilmArray Respiratory Panel (BioFire Diagnostics, Salt Lake City, UT, USA), were performed on the induced sputum samples collected after nebulization. This assay identified hMPV as the sole pathogen, excluding bacterial and other viral coinfections. Unlike nasopharyngeal swabs, which are used exclusively for SARS-CoV-2 testing, induced sputum samples provide a more representative evaluation of lower respiratory secretions. Molecular testing primarily detects specific bacterial pathogens, including atypical or subclinical infections. Globally, multiplex RT-PCR is recognized as a standard diagnostic tool; however, its adoption in regions with limited access to advanced molecular diagnostics, such as Brazil, significantly advances clinical practice. This technology is critical for addressing the diagnostic challenges of overlapping clinical presentations and the prevalence of viral and bacterial coinfections.

These limitations, combined with the severity of the patient’s clinical condition and the potential risk of bacterial superinfection—frequently observed in cases of severe viral pneumonia, as highlighted by Nagasawa et al. [14]—justified the empirical use and continuation of antibiotics.

This case underscores the diagnostic challenges of distinguishing between viral and bacterial pneumonia, even with advanced molecular tools. While molecular diagnostics significantly enhances pathogen identification and guides clinical decision-making, their results must be interpreted alongside clinical and radiological findings to ensure optimal management [15,16]. Reverse transcription PCR, recognized as the gold standard for hMPV diagnosis, is characterized by its high sensitivity and specificity [17]. The PERCH study [5] further demonstrated the value of advanced molecular diagnostics in revealing underdiagnosed viral pathogens like hMPV and RSV, even in high-burden pneumonia settings, highlighting discrepancies between blood cultures and multiplex PCR results, with the latter offering greater sensitivity.

This case reinforces the importance of integrating molecular diagnostics into routine clinical workflows for CAP, particularly for patients with atypical presentations. Thrombocytopenia observed in this patient is a frequent finding in severe viral infections and is associated with disease severity. However, it is a non-specific marker that can also occur in critically ill patients with bacterial or other systemic infections, reflecting the complex interplay between inflammation, coagulation, and endothelial damage. Studies have demonstrated that thrombocytopenia correlates with worse outcomes in critically ill patients and serves as an important, albeit non-specific, marker of disease severity [18,19,20].

#### 3.2.2. Clinical-Radiological Correlation

The radiological findings, characterized by diffuse ground-glass opacities and consolidations predominantly in the lower lobes, are consistent with the typical imaging pattern observed in viral pneumonia [21]. These radiological features overlap with those observed in other viral infections, such as SARS-CoV-2 and RSV, complicating diagnostic differentiation. However, the absence of bacterial consolidation patterns and low procalcitonin levels substantiated a viral etiology without bacterial superinfection [2]. This case highlights the importance of integrating radiological, clinical, and molecular data for accurate diagnostic assessment during seasonal respiratory surges.

### 3.3. Epidemiological and Public Health Implications

The underdiagnosed burden of hMPV in adult populations is evident in epidemiological studies [12,15,17,22,23]. Incorporating hMPV testing into CAP diagnostic workflows could significantly improve pathogen identification, particularly during seasonal surges in respiratory viruses. The absence of FDA-approved antivirals or vaccines against hMPV represents a critical gap in respiratory disease management. Promising advancements in vaccine development, including virus-like particles, RNA-based platforms, and stabilized fusion proteins, offer hope for effective prevention strategies against human metapneumovirus (hMPV) and related viruses such as respiratory syncytial virus (RSV)**.** For example, investigational bivalent vaccines like IVX-A12, which are currently in phase II trials and designed to target both hMPV and RSV, highlight ongoing efforts to fill critical gaps in preventing and managing respiratory infections. These efforts underscore the importance of sustained investment in vaccine research to mitigate the global burden of these diseases [24,25].

While this case highlights the clinical impact of hMPV, it is important to note that the contribution of a single case report lies primarily in raising awareness. The overlapping clinical presentation of hMPV with other respiratory viruses such as SARS-CoV-2, influenza, and RSV underscores the need for an accurate differential diagnosis of severe respiratory infections. This case reinforces the critical role of rapid molecular diagnostics in identifying under-recognized viral pathogens, thereby enabling timely and precise clinical decision-making.

Recent epidemiological studies [14] have shown that hMPV is a major contributor to CAP in individuals over 60 years of age, with many cases requiring hospitalization due to severe symptoms. These findings further emphasize the importance of including hMPV in the differential diagnoses of severe respiratory infections and support the need for comprehensive molecular diagnostics to improve pathogen identification and guide clinical management.

This case underscores the necessity of integrating molecular diagnostics into routine workflows by aligning them with these observations. Such measures would enhance early detection, optimize resource allocation, and improve patient outcomes, particularly in vulnerable populations like older adults. Addressing the global impact of hMPV will require not only advancements in diagnostics and therapeutics but also equitable implementation of these innovations to reduce morbidity and mortality associated with hMPV-related illnesses.

### 3.4. Transmission and Clinical Presentation

hMPV spreads primarily through respiratory secretions from coughing or sneezing, close personal contact (e.g., handshakes), touching contaminated surfaces, and subsequently touching the eyes, nose, or mouth. These modes of transmission pose particular risks in healthcare settings where proper infection control measures, such as hand hygiene and surface disinfection, are essential to prevent nosocomial spread [26]. The virus remains transmissible for up to one week after symptom onset, necessitating vigilance among healthcare workers. Clinically, hMPV presents with symptoms similar to those of other respiratory viruses. Upper respiratory symptoms include cough, rhinorrhea, sore throat, and fever, while lower respiratory symptoms can manifest as wheezing, dyspnea, and hypoxia. Despite the molecular confirmation of viral etiology, empirical treatment with antibiotics and corticosteroids remains common, reflecting persistent diagnostic challenges [21].

The overlap in clinical and radiological findings between hMPV and other pathogens, such as SARS-CoV-2 and RSV, underscores the need for precise diagnostics, particularly in the post-pandemic era.

### 3.5. Figure Legend

This graphical abstract outlines the diagnostic challenges and management strategies for a case of severe hMPV pneumonia in an immunocompetent adult. The “Clinical Presentation” section highlights key imaging findings, including ground-glass opacities and consolidations, and confirms the diagnosis via multiplex PCR. The “Signals & Symptoms” section summarizes clinical manifestations, such as thick yellowish-brown expectoration and respiratory distress, alongside a negative SARS-CoV-2 test result. The “Management” section depicts supportive care measures, including intravenous antibiotics, hydration, nebulization, and post-discharge monitoring. Finally, the “Outcome” section underscores the patient’s complete recovery within 14 days, emphasizing the efficacy of a comprehensive diagnostic and therapeutic approach.

Graphical Abstract: Illustration created using the EdrawMax (Wondershare) software. Source: Edrawsoft (Version 14.1.0; Release date: 12 November 2024).

## 4. Conclusions

This case highlights the clinical impact of hMPV-related pneumonia in immunocompetent adults and underscores the importance of rapid molecular diagnostics, such as multiplex RT-PCR, to enable precise pathogen identification and guide clinical management. These findings emphasize the importance of heightened clinical awareness of hMPV, particularly in older adults and during respiratory virus seasonality. Addressing the global burden of hMPV requires ongoing investment in diagnostics and therapies to improve outcomes and reduce the associated morbidity.

## Figures and Tables

**Figure 1 microorganisms-13-00073-f001:**
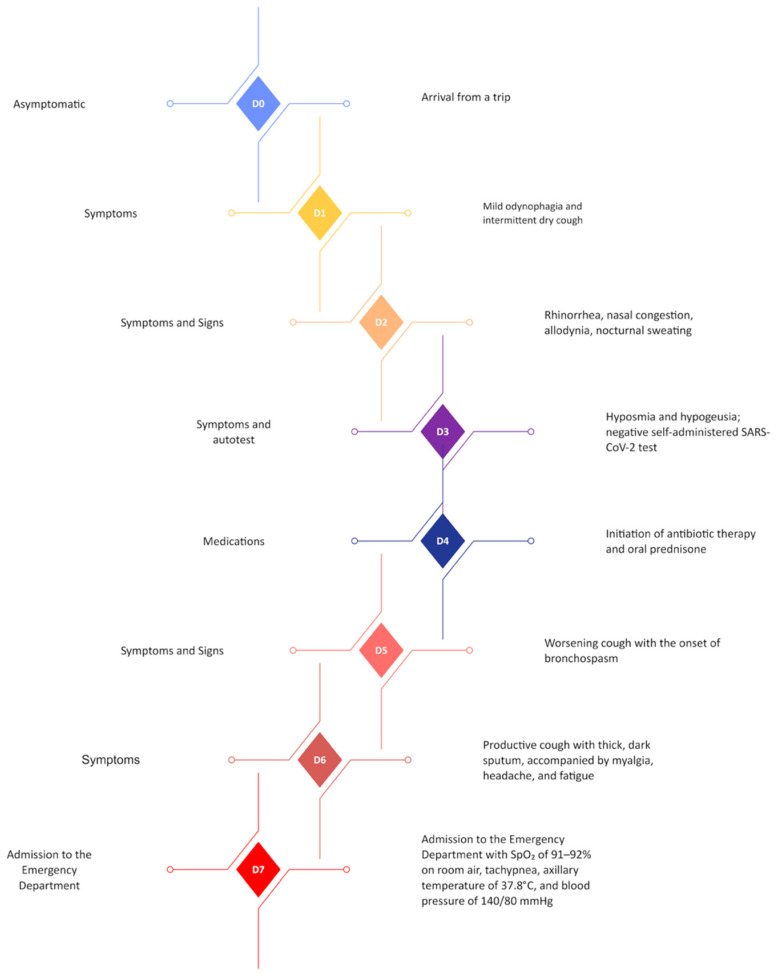
Timeline of symptom progression and clinical events leading to hospital admission. The diagram illustrates the patient’s clinical course from symptom onset (Day 0) to hospital admission (Day 7). The patient’s recent domestic trip to a resort in Angra dos Reis, located two hours by car from Rio de Janeiro, was uneventful and posed no risk of tropical or endemic infections. Key events highlight the progression from mild respiratory symptoms to severe disease, marked by hypoxia and fatigue, necessitating hospitalization. Therapeutic interventions and diagnostic milestones are also described.

**Figure 2 microorganisms-13-00073-f002:**
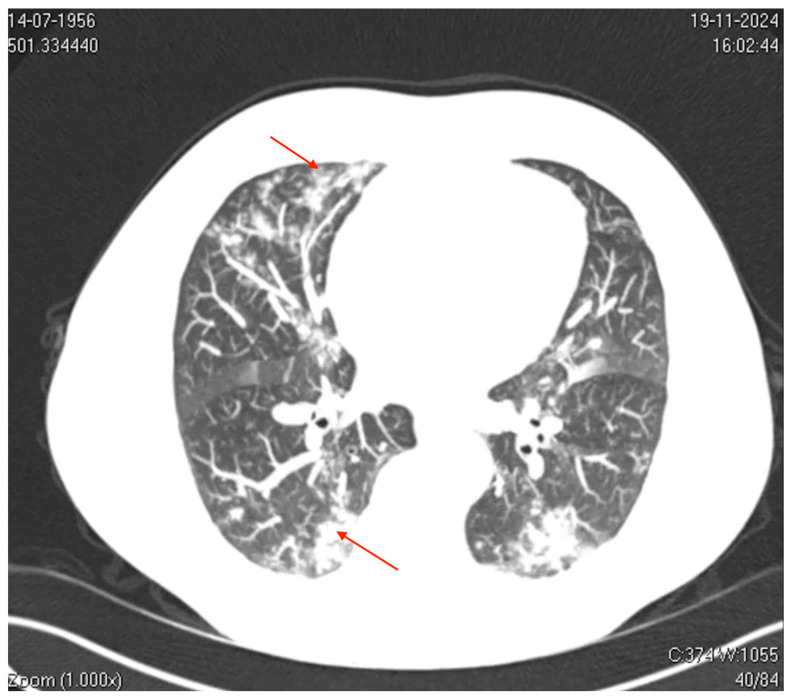
Axial computed tomography (CT) image of the thorax. The image shows bilateral ground-glass opacities with a reticular pattern interspersed with focal areas of consolidation and peripheral bronchiolar filling. The red arrows highlight specific areas of consolidation surrounded by ground-glass opacities, which are more prominent in the peripheral and lower lung regions. These findings suggest a diffuse inflammatory or infectious process involving the pulmonary interstitial and peripheral airways (acinus and bronchioles), which is consistent with viral pneumonia.

**Figure 3 microorganisms-13-00073-f003:**
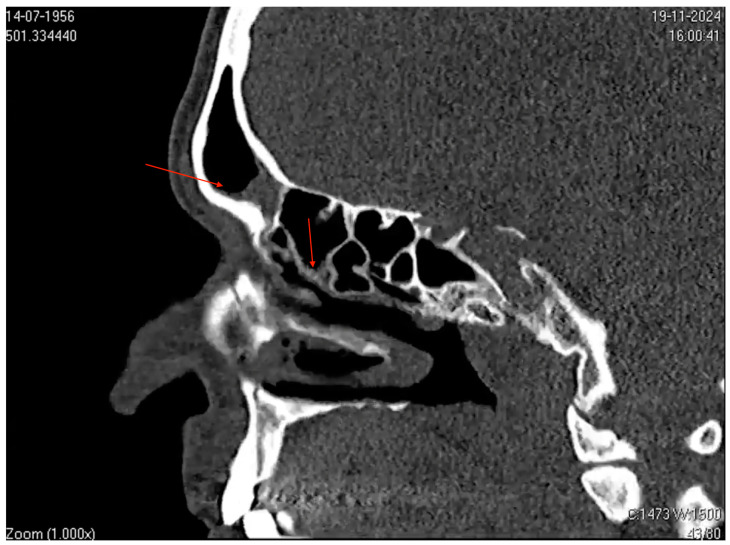
Computed tomography (CT) image of the paranasal sinuses in the sagittal plane. The image shows mucosal thickening and partial opacification in the frontal sinus (highlighted by the superior red arrow) and ethmoid air cells (highlighted by the inferior red arrow). Narrowing of the sinus drainage pathways is also evident. These findings are consistent with those of acute and subacute sinusitis.

**Table 1 microorganisms-13-00073-t001:** Molecular diagnostic results obtained through multiplex RT-PCR and a molecular panel.

Pathogen Category	Detected Pathogen	Result	Methodology
Viral Pathogens	Human Metapneumovirus	Positive	Multiplex RT-PCR
	Influenza A, B	Not Detected	
	RSV	Not Detected	
	Adenovirus	Not Detected	
Bacterial Pathogens	Multiple (e.g., Klebsiella pneumoniae, Staphylococcus aureus)	Not Detected	Multiplex RT-PCR
Atypical Bacteria	Legionella pneumophila, Mycoplasma pneumoniae	Not Detected	Multiplex RT-PCR
Resistance Genes	CTX-M, KPC, NDM	Not Detected	Molecular Panel

## Data Availability

The clinical data supporting the findings of this case report are derived from the patient’s medical records, which are under the custody of Hospital Copa Star, Rio de Janeiro, Brazil. Due to confidentiality agreements and institutional policies, these records are not publicly available. Specific information may be accessible upon reasonable request with the approval of the hospital’s ethics committee. For inquiries, please contact Rubens C. Costa-Filho at rubens1956@gmail.com.

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
