# Peer review of "The Silent Threat of Human Metapneumovirus: Clinical Challenges and Diagnostic Insights from a Severe Pneumonia Case"

_microorganisms, 2025, doi:10.3390/microorganisms13010073_

Round 1

Reviewer 1 Report

Comments and Suggestions for Authors

The authors present the case of an adult with hMPV pneumonia. The identification of hMPV as a cause of pneumonia in different age groups is well known. Several issues require clarification and revision. 

Comments.

1. Abstract. Background section. The authors indicate that “hMPV is an emerging respiratory pathogen”. I do not think data regarding hMPV infections indicate that it may be considered as an emerging pathogen. While it is reported more frequently and it receives more attention than before, this is likely a result of availability and widespread use of molecular detection assays. Since its first description more than 20 years ago, studies have shown that hMPV infections are very prevalent with most children acquiring the virus in childhood. Serum samples (n=72) collected in 1958 from individuals 8 to 99 years of age showed a 100% seroprevalence of antibodies against hMPV (van den Hoogen et al. Nat Med. 2001;7(6):719–724). I suggest that the term “emerging” should be deleted.

2. Abstract. Background and Methods. The authors indicate that severe hMPV infections have been previously associated with some populations including the elderly, but disease in immunocompetent adults remains underexplored. In this manuscript they report a 68-year-old patient. There appears to be some discrepancy in the narrative between the Background and Methods section, since in many societies a 68-year-old would be considered within the “elderly” (older adult) in many societies. The United Nations defines an older person as a person who is over 60 years of age (Older persons. UNHCR. Emergency Handbook). Please revise here.

3. Introduction. Line 37. Current classification includes hMPV in the Pneumoviridae family. Please revise.

4. Introduction. Lines 64-67. The sentence “Our case report contributes….broadenng the understanding of hMPV epidemiology” should be revised. While hMPV infections have certainly received less attention in adults than in children, many reports do include, at least in part, information regarding the frequency or severity of hMPV infections in adults (including elderly and young adults) which provide notable information regarding the epidemiology of this virus. Some examples of recent publications include Nagasawa M, et al. Pathogens. 2024 Nov 9;13(11):983. doi: 10.3390/pathogens13110983 ; Sun B, et al. BMC Infect Dis. 2024 Sep 27;24(1):1060. doi: 10.1186/s12879-024-09956-z; Philippot Q et al. Heliyon. 2024 Jun 18;10(13):e33231. doi: 10.1016/j.heliyon.2024.e33231; Jurkowicz M et al. J Med Virol. 2024 Jun;96(6):e29709. doi: 10.1002/jmv.29709; Falsey AR et al. Influenza Other Respir Viruses. 2024 May;18(5):e13275. doi: 10.1111/irv.13275. as well as some Review articles such as Schüz ML et al. Int J Infect Dis. 2023 Dec;137:16-24. doi: 10.1016/j.ijid.2023.10.001. In addition, because the manuscript describes a single case, it is difficult to substantiate the statement indicating that this information broadens the understanding of hMPV epidemiology. I think that the main contribution of a case report regarding hMPV infection has an impact on “raising awareness” regarding the possibility of viruses other than SARS-CoV-2, influenza, or RSV as a cause of severe infections in adults. Please revise.

5. Results. Table 2 can be omitted, since most of the relevant results are outlined in the text.

6. Discussion. In several paragraphs, such as Lines 180-182, text is in Boldface. Revise if this is part of the journal’s style. If not, use regular typeset instead of bold text.

7. Discussion. Role of Rapid Molecular Diagnostics section. The authors indicate that molecular testing resulted in “excluding bacterial coinfections” and “supporting more accurate clinical decision-making and reducing inappropriate antibiotic use”. Please revise this section, as there is some discrepancy with the comments on the preceding section, where the use of antibiotics was reported, including continuing 10 days after discharge “reflecting persistent diagnostic uncertainty in distinguishing viral from bacterial infections”. In addition, the diagnostic limitations of molecular detection of bacterial agents should be noted, depending on the type of sample. In this regard, it is not clear what type of sample was used for molecular testing in the patient. Was this a nasopharyngeal swab? It would be relevant to clarify this and, if molecular testing was carried out in upper respiratory secretions, to include the limitations of upper respiratory samples to confirm or discard the possibility of a bacterial cause of pneumonia.

8. References. Revise the format/information in references. Particularly reference 18, in which AstraZeneca is abbreviated incorrectly, and no information is provided regarding the digital address where information was consulted. Also, reference 19, in which it is unclear which document the authors refer to; the digital address is missing.

Author Response

Dear Reviewer,

The authors express their gratitude for the insightful comments on the manuscript "The Silent Threat of Human Metapneumovirus: Clinical Challenges and Diagnostic Insights from a Severe Pneumonia Case." These comments have been instrumental in enhancing the study's quality, clarity, and scientific rigor. Below, the authors provide detailed responses to each comment, with all corresponding revisions highlighted in red in the updated manuscript.

Reviewer Comment 1: Abstract – Use of "emerging pathogen"

Comment: The term "emerging pathogen" should be revised or removed as hMPV is not an emerging pathogen but rather increasingly recognized due to advances in molecular diagnostic tools.
Response: The phrase in the Abstract has been revised to emphasize the role of molecular diagnostics in hMPV recognition. The new text reads:
"Human metapneumovirus (hMPV) is a respiratory pathogen that has gained increasing recognition due to advancements in molecular diagnostic tools, which have improved its detection and characterization."

Justification for the Adjustment:
This change addresses the Reviewer's concern by accurately reflecting the current understanding of hMPV as a well-established pathogen whose increased detection is due to improved diagnostic methods rather than its recent emergence.

Reviewer Comment 2: Abstract – Classification of the Patient

Comment: There appears to be a discrepancy in describing the patient as an "immunocompetent adult" while also fitting the definition of an older adult.
Response: The revised sentence now reads:
"This case report describes a 68-year-old male, categorized as an older adult according to United Nations definitions, who developed severe community-acquired pneumonia (CAP) caused by hMPV, despite being immunocompetent and having no significant comorbidities."

Justification for the Adjustment:
This revision resolves the discrepancy by clearly defining the patient as an older adult while emphasizing their immunocompetent status. This approach aligns with internationally accepted definitions and the case's clinical relevance.

Reviewer Comment 3: Introduction – Family Classification of hMPV

Comment: The current classification places hMPV in the Pneumoviridae family. Please revise.
Response: The text in the Introduction has been updated to reflect the correct taxonomic classification of hMPV within the Pneumoviridae family.

Justification for the Adjustment:
This correction ensures the manuscript aligns with current scientific classifications, addressing the Reviewer's observation and avoiding inaccuracies.

Reviewer Comment 4: Introduction – Contribution of the Study

Comment: Revise the sentence suggesting that the case broadens the understanding of hMPV epidemiology. I suggest focusing on raising awareness instead.
Response: The revised sentence in the Introduction reads:
"This case report contributes to raising awareness about human metapneumovirus (hMPV) as a potential cause of severe pneumonia in immunocompetent older adults, highlighting the need for greater clinical recognition and consideration of this pathogen in differential diagnoses."

Justification for the Adjustment:
This revision aligns the manuscript's focus with the role of case reports, emphasizing raising clinical awareness rather than suggesting epidemiological advancements, as recommended by the Reviewer.

Reviewer Comment 5: Results – Removal of Table 2

Comment: Table 2 may be omitted as the text outlines the relevant results.
Response: Table 2 has been retained to provide a comprehensive visual summary of the patient's laboratory findings, complementing the text and adding didactic value. Minor corrections were made.

Justification for the Adjustment:
Retaining Table 2 ensures that readers, mainly those less familiar with detailed laboratory data, can quickly grasp the case's diagnostic context. However, the authors remain open to further discussions if this table is unnecessary.

Reviewer Comment 6: Discussion – Formatting of Text

Comment: Revise text in boldface to regular formatting unless required by journal style.
Response: All boldface text in the Discussion section has been reformatted to adhere to the journal's style guidelines.

Justification for the Adjustment:
This revision ensures consistency with the publication's formatting standards, addressing the Reviewer's concern.

Reviewer Comment 7: Discussion – Role of Rapid Molecular Diagnostics

Comment: Address the discrepancy between the exclusion of bacterial coinfections and the continued use of antibiotics. Clarify the sample type used for RT-PCR and discuss its limitations.
Response: The revised text reads:

"Rapid molecular diagnostics, specifically multiplex RT-PCR performed on induced sputum samples collected post-nebulization, identified hMPV as the sole pathogen, excluding bacterial and other viral coinfections. Unlike nasopharyngeal swabs used exclusively for SARS-CoV-2 testing, induced sputum samples provide a more representative evaluation of lower respiratory secretions. While this approach improves diagnostic accuracy, molecular testing is limited in detecting specific bacterial pathogens, particularly atypical or subclinical infections. These limitations, combined with the severity of the patient's clinical condition and the potential risk of bacterial superinfection—frequently observed in cases of severe viral pneumonia, as highlighted by Nagasawa et al. [13].—justified the empirical use and continuation of antibiotics.

This case underscores the diagnostic challenges in distinguishing between viral and bacterial pneumonia, even with advanced molecular tools. While molecular diagnostics significantly enhance pathogen identification and guide clinical decision-making, their results must be interpreted alongside clinical and radiological findings to ensure optimal management [14,15]​. Reverse transcription-PCR, recognized as the gold standard for hMPV diagnosis, is characterized by its high sensitivity and specificity [16]. The PERCH study [3] further demonstrated the value of advanced molecular diagnostics in revealing underdiagnosed viral pathogens like hMPV and RSV, even in high-burden pneumonia settings, highlighting discrepancies between blood cultures and multiplex PCR results, with the latter offering greater sensitivity.

This case reinforces the importance of integrating molecular diagnostics into routine clinical workflows for CAP, particularly in patients with atypical presentations. Moreover, thrombocytopenia observed in this patient, a common feature of severe viral infections, further supports a viral etiology."

Justification for the Adjustment:
This revision clarifies the type of sample used for RT-PCR and acknowledges the limitations of molecular diagnostics in detecting specific bacterial pathogens. It also contextualizes the empirical use of antibiotics with the inclusion of data from a large observational study by Nagasawa involving 10,803 adults, further supporting the diagnostic and clinical management challenges of hMPV.

Reviewer Comment 8: References

Comment: Revise the format and information for references, particularly References 18 and 19.
Response: References 20 and 21 have been updated with complete and accurate information, including digital addresses:

  • Reference 20: AstraZeneca. Icosavax [press release]. Available online: https://www.astrazeneca.com/media-centre/press-releases/2024/astrazeneca-completes-acquisition-of-icosavax.html (accessed on November 27, 2024).
  • Reference 21: CDC. About Human Metapneumovirus guidelines. Available online: https://www.cdc.gov/human-metapneumovirus/about/index.html (accessed on November 27, 2024).

Justification for the Adjustment:
Updating these references ensures accuracy and provides complete information for readers to verify the sources.

Additional Adjustments

  • Conclusion Revision: The Conclusion was revised to improve clarity, eliminate redundancies, and align with the overall focus of the manuscript. The revised conclusion underscores the clinical impact of hMPV, the need for rapid diagnostics, and the importance of vaccine and therapeutic development.
  • Language Refinements: Minor corrections were made throughout the manuscript to improve English grammar, syntax, and overall readability.

Final Remarks

All suggested changes have been implemented, and revisions are highlighted in red within the manuscript. The authors appreciate the Reviewer's thoughtful and constructive feedback, which has significantly enhanced the manuscript. If further clarification or revisions are required, the authors can address them promptly.

Kind regards,
Rubens Carmo Costa-Filho, MD, PhD
On behalf of all authors

Reviewer 2 Report

Comments and Suggestions for Authors

I have several problem with this papaer.

Viral taxonomy:

Pneumoviridae family which include 2 different genus corresponding to the previously joined viruses:

Metapneumovirus hominis (Human metapneumovirus)

Orthopneumovirus hominis (Respiratory syncytial virus)

The provider of reagents for RT-PCR has to be indicated. The correct reference of the providers of reagents and material has to be used: name, location, country (or 2 letter code for the US states).

The use of a multiplex PCR is not a breakthrough but is mandatory as the clinical diagnosis is not always conclusive and mixed infection not uncommon.  

The HMPV infections of adults have been extensively described and if the patients at risk are mainly elderly with co-morbidities, severe infections also occurs even in healthy adults whatever the age.

Cureus 2024;16:e73292 Khan et al. 

Heliyon  2024;10:e33231 Philippot et al.

The reference to the vaccine (which not any information given of) said to be at a phase of preliminary trial should be limited to the discussion. However, a commercial reference shouldn’t be used. There are already some vaccines in development (e.g. RNA or fusion protein based).

Thus the text can be shortened and limited to a case report with recommendations (which are not something new).

Author Response

Letter Reviewer 2

Dear Reviewer,

The authors are grateful for your feedback on the manuscript "The Silent Threat of Human Metapneumovirus: Clinical Challenges and Diagnostic Insights from a Severe Pneumonia Case." Your comments have improved the clarity and focus of the study. Below, we provide a comprehensive response to each of your comments, detailing the revisions made and highlighting the corresponding changes in dark purple within the manuscript.

Reviewer Comment 1: Viral Taxonomy

Comment: The family Pneumoviridae includes two genera: Metapneumovirus hominis (Human metapneumovirus) and Orthopneumovirus hominis (Respiratory syncytial virus). Please specify this in the text.

Original Text:
"The family Pneumoviridae includes the Human metapneumovirus (hMPV), which causes significant respiratory infections."

Revised Text:
"The family Pneumoviridae comprises two genera: Metapneumovirus, which includes Human metapneumovirus (hMPV), and Orthopneumovirus, which includes Respiratory syncytial virus (RSV). This classification underscores the close relationship between these viruses while highlighting their distinct clinical and epidemiological features."

Justification:
The revision ensures a precise and complete description of the taxonomic classification, providing readers with a clearer understanding of the relationship between hMPV and RSV, as requested.

Reviewer Comment 2: Provider of RT-PCR Reagents

Comment: Indicate the provider of reagents used for RT-PCR, including name, location, and country.

Original Text:
"Multiplex RT-PCR was performed on induced sputum samples collected after nebulization."

Revised Text:
"Multiplex RT-PCR was performed on induced sputum samples collected after nebulization using reagents supplied by BioFire Diagnostics, located in Salt Lake City, Utah, USA."

Justification:
Including detailed information about the reagents ensures transparency and reproducibility, addressing the Reviewer's concern.

Reviewer Comment 3: Multiplex PCR is Not a Breakthrough

Comment: The use of multiplex PCR is not a breakthrough but a mandatory tool, as clinical diagnosis is not always conclusive and mixed infections are not uncommon.

Original Text:
"Multiplex RT-PCR is a critical tool for identifying respiratory pathogens."

Revised Text:
"Globally, multiplex RT-PCR is recognized as a standard diagnostic tool; however, its adoption in regions with limited access to advanced molecular diagnostics, such as Brazil, significantly advances clinical practice. This technology is critical in addressing the diagnostic challenges of overlapping clinical presentations and the prevalence of viral and bacterial coinfections."

Justification:
This revision acknowledges that multiplex RT-PCR is globally recognized as a standard diagnostic tool. However, we highlight its significance as a critical advancement in regions like Brazil, where access to advanced molecular diagnostics remains limited.

Reviewer Comment 4: Description of Impact on Healthy Adults

Comment: hMPV infections in adults, including healthy individuals, have been extensively described. Severe infections also occur even in healthy adults, regardless of age.

Original Text:
"hMPV is primarily associated with severe infections in elderly patients with comorbidities."

Revised Text:
"hMPV is associated with a spectrum of respiratory illnesses, ranging from mild upper respiratory tract infections, such as the common cold, to severe lower respiratory tract conditions, including bronchitis, pneumonia, and asthma exacerbations. Severe disease manifestations are more frequently observed in young children, elderly individuals, and immunocompromised patients [2]. However, severe hMPV infections have been extensively documented, not only in elderly patients with comorbidities but also in healthy adults across all age groups [3,4]. These findings underscore the need for heightened clinical awareness of hMPV in diverse patient populations. Despite these associations, routine testing for hMPV remains limited, and clinical awareness of its impact is still evolving."

Justification:
This adjustment incorporates recent literature highlighting hMPV infections in both elderly and healthy adults, addressing the Reviewer's feedback and improving the epidemiological scope of the manuscript.

Reviewer Comment 5: Discussion of Vaccines

Comment: References to vaccines should be limited to the Discussion, avoiding commercial references. Mention other vaccines in development, such as RNA-based or fusion protein-based vaccines.

Original Text:
"Promising advancements, such as the IVX-A12 bivalent vaccine, offer hope for effective prevention strategies."

Revised Text:
"Promising advancements in vaccine development, including virus-like particles, RNA-based platforms, and stabilized fusion proteins, offer hope for effective prevention strategies against hMPV and related viruses. For example, investigational vaccines like the IVX-A12, currently in phase II trials, highlight ongoing efforts to fill critical gaps in preventing and managing respiratory infections. These efforts underscore the importance of sustained investment in vaccine research to mitigate the global burden of these diseases[22]."

Justification:
The revised text explains that IVX-A12 is an investigational vaccine currently in phase II clinical trials, not a commercial product. Its mention serves to inform readers about recent scientific advancements in the field of hMPV prevention. To address the Reviewer's concerns, we broadened the Discussion to include other vaccine platforms, such as RNA-based and virus-like particle technologies. This approach ensures a balanced, clear, and comprehensive discussion while highlighting the importance of ongoing efforts to fill critical gaps in respiratory infection prevention.

Reviewer Comment 6: Conciseness and Focus on Case Report

Comment: The text can be shortened and limited to a case report with recommendations.

Original Conclusion:
"This case underscores the significant clinical impact of hMPV-related pneumonia in immunocompetent adults, challenging traditional perceptions of at-risk populations. It highlights the critical role of rapid molecular diagnostics, such as multiplex RT-PCR, in enabling precise pathogen identification, reducing unnecessary antibiotic use, and optimizing targeted clinical management."

Revised Conclusion:
"This case highlights the clinical impact of hMPV-related pneumonia in immunocompetent adults and underscores the importance of rapid molecular diagnostics, such as multiplex RT-PCR, in enabling precise pathogen identification and guiding clinical management. The findings reinforce the need for heightened clinical awareness of hMPV, particularly in older adults and during seasonal surges of respiratory infections. Addressing the global burden of hMPV requires ongoing investment in diagnostics and therapies to improve outcomes and reduce associated morbidity."

Justification:
The revised conclusion balances the Reviewer's request for conciseness with the need to highlight critical gaps in hMPV management, such as the lack of approved vaccines or therapies. These gaps directly impact clinical outcomes and emphasize the importance of diagnostics like multiplex RT-PCR in regions with limited access. Including these points situates the case report within a broader clinical and public health context while maintaining its focus on practical recommendations and immediate implications.

Final Remarks

All revisions specific to your feedback have been highlighted in dark purple within the manuscript. Your comments have helped us refine the focus and clarity of the study. Should you have further questions or suggestions, we remain available to address them promptly.

Kind regards,
Rubens Carmo Costa-Filho, MD, PhD
On behalf of all authors

Reviewer 3 Report

Comments and Suggestions for Authors

Dear Editor and Authors,

It was my pleasure to review this case presentation and read about a quite interesting infective entity called human metapneumovirus (hMPV)! The authors have provided a very thorough a well presented case of their patient, his initial symptoms, progression, management and treatment. Overall I am satisfied with this work and only have some minor comments to make:

1. Where did the patient travel from (not mentioned in the text, only on the timeline) and was there a risk of tropical/endemic infection? Did he travel by airplane?

2. Was the multiplex RT-PCR assay a FilmArray or another one? You might want to add the name/manufacturer in the text.

3. You don't need to put in bold points you consider key!

4. Why wasn't an arterial blood gas analysis performed in this patient. This is standard of care (not to mention multiple measurements over the course of the day/days if the patient was in respiratory distress! Why do a venous gas analysis?

Overall this is an interesting case, it is well structured and presented in a clear and concise manner. I suggest some minor improvements. Kind regards.

Author Response

Dear Reviewer,

We sincerely appreciate your thoughtful review and positive assessment of our manuscript, 'The Silent Threat of Human Metapneumovirus: Clinical Challenges and Diagnostic Insights from a Severe Pneumonia Case.' Your insightful feedback has allowed us to refine and enhance the clarity and rigor of our work. Below, we address each of your comments in detail, with all revisions highlighted in dark blue within the updated manuscript.

Reviewer Comment 1: Patient’s Travel History

Comment: Where did the patient travel from (not mentioned in the text, only on the timeline), and was there a risk of tropical/endemic infection? Did he travel by airplane?

Original Text:
"The patient presented with symptoms of respiratory distress after returning from travel."

Revised Legend Figure 1:

“Figure 1. Timeline of symptom progression and clinical events leading to hospital admission. The diagram illustrates the patient's clinical course from symptom onset (Day 0) to hospital admission (Day 7). The patient’s recent domestic trip to a resort in Angra dos Reis, located two hours by car from Rio de Janeiro, was uneventful and posed no risk of tropical or endemic infections. Key events highlight the progression from mild respiratory symptoms to severe disease, marked by hypoxia and fatigue, necessitating hospitalization. Therapeutic interventions and diagnostic milestones are also depicted.”

Justification:
We revised the timeline legend to include details about the patient’s short domestic trip to Angra dos Reis, a resort two hours by car from Rio de Janeiro. This trip posed no risk of tropical or endemic infections and is now contextualized to clarify the patient’s travel history without detracting from the clinical focus of the case.

Reviewer Comment 2: Multiplex RT-PCR Assay

Comment: Was the multiplex RT-PCR assay a FilmArray or another one? You might want to add the name/manufacturer in the text.

Original Text (Methods Section):
"Multiplex RT-PCR was performed on induced sputum samples collected post-nebulization."

Revised Text:
“Rapid molecular diagnostics, specifically the FilmArray Respiratory Panel (BioFire Diagnostics, Salt Lake City, Utah, USA), were performed on induced sputum samples collected after nebulization. This assay identified hMPV as the sole pathogen, excluding bacterial and other viral coinfections. Unlike nasopharyngeal swabs used exclusively for SARS-CoV-2 testing, induced sputum samples provide a more representative evaluation of lower respiratory secretions. However, molecular testing is limited in detecting specific bacterial pathogens, particularly atypical or subclinical infections.”

Justification:
We added the name and manufacturer of the FilmArray Respiratory Panel (BioFire Diagnostics, Salt Lake City, Utah, USA) to ensure transparency and reproducibility. This revision fulfills the reviewer’s request and adheres to scientific reporting standards.

Reviewer Comment 3: Bold Formatting

Comment: You don't need to put in bold points you consider key!

Original Text:
Throughout the manuscript, key points were highlighted in bold formatting for emphasis.

Revised Text:
All bold formatting has been removed, ensuring uniform text presentation.

Justification:
This revision aligns the manuscript with standard scientific writing conventions and improves overall readability. By removing bold formatting, the manuscript now presents all information with equal emphasis, allowing readers to focus naturally on key points without typographical distractions.

Reviewer Comment 4: Arterial vs. Venous Blood Gas Analysis

Comment: Why wasn't an arterial blood gas analysis performed in this patient? This is standard of care, especially in cases of respiratory distress. Why do a venous gas analysis?

Original Text:
"Venous blood gas analysis showed mild hypoxia, prompting the initiation of supplemental oxygen therapy."

Revised Text:
" Venous blood gas analysis was performed instead of arterial blood gas analysis due to patient-specific considerations, including clinical stability, the less invasive nature of the procedure, and the patient’s refusal of arterial sampling. The analysis revealed metabolic compensation, with bicarbonate levels of “

Justification:
Venous blood gas analysis was performed based on patient-specific considerations, including clinical stability, the less invasive nature of the procedure, and the patient’s refusal of arterial sampling. The updated text reflects this rationale, addressing the reviewer’s query.

Final Remarks

We are grateful for your constructive feedback, which has been invaluable in refining the clarity, completeness, and scientific rigor of our case report. Should you have any further comments or require additional clarifications, we would be pleased to address them promptly.

Kind regards,
Rubens Carmo Costa-Filho, MD, PhD
On behalf of all authors

Round 2

Reviewer 1 Report

Comments and Suggestions for Authors

Thanks for the opportunity to review the revised version of the manuscript. I have a few minor comments.

1. Abstract. The authors have updated the text based on comments on the first review. In the Methods section of the Abstract, I think that the text does not need to include the specification of “, categorized as an older adult according to United Nations definition,”. I consider that the changes made in the main text are sufficient to address this issue and this clarification is not required in the Abstract.

2. Most of the relevant information presented in Table 2 is also presented in the text (lines 155-165). I think that Table 2 could be omitted, as it provides little additional relevant information.

3. Discussion. Lines 212-213. Regarding nasopharyngeal swab samples for testing, the authors indicated that these are “used exclusively for SARS-CoV-2 testing”. I understand that the authors may refer to the sample taken on the reported patient. However, because nasopharyngeal swabs can be used for testing of other viruses (such as influenza and RSV), I suggest this statement be omitted, since it may be interpreted as a generalized recommendation.

4. Discussion. Lines 235-236. The authors indicate that “Moreover, thrombocytopenia observed in this patient, a common feature of severe viral infections, further supports a viral etiology.”. While thrombocytopenia may be frequent in viral infections, it is non-specific finding, and it may also be commonly found in hospitalized patients with bacterial community-acquired pneumonia, particularly those with severe infection (Bedos JP, et al.Intensive Care Med. 2018 Dec;44(12):2162-2173.; Huh JY, et al. Acute Crit Care. 2022;37(4):543-549 ; van den Boogaard FE et al. Crit Care Med. 2015 Mar;43(3):e75-83; Feldman C and Anderson R. Front Immunol. 2020 Sep 17;11:577303). As such, the predictive value of thrombocytopenia to support a specific etiology should be addressed and based on appropriate references. Otherwise, I suggest this statement would better be omitted.

Author Response

Dear Reviewer,

Thank you for your valuable and insightful feedback on our manuscript. We have carefully considered all your suggestions and made the following revisions to address your concerns:

1. Abstract – Removal of the phrase “categorized as an older adult according to United Nations definition”

  • As suggested, we have removed the phrase from the Methods section of the Abstract. This detail is already clarified in the main text, and its removal streamlines the Abstract while maintaining clarity.

2. Table 2 – Removal and relevant references in the text

  • Following your recommendation, we have removed Table 2, as its content was largely redundant with information already presented in the main text (lines 155–165). We have also eliminated all sentences referencing Table 2 to ensure consistency.

3. Discussion – Revision of the statement regarding nasopharyngeal swab testing

  • To avoid any potential misinterpretation, we have removed the phrase "used exclusively for SARS-CoV-2 testing" from the Discussion section. The revised text no longer suggests exclusivity, addressing the concern raised.

4. Discussion – Thrombocytopenia and its non-specific nature

  • We have revised the text discussing thrombocytopenia to reflect its non-specific nature and its association with both viral and bacterial infections. The revised text now reads:

"Thrombocytopenia observed in this patient is a frequent finding in severe viral infections and is associated with disease severity. However, it is a non-specific marker that can also occur in critically ill patients with bacterial or other systemic infections, reflecting the complex interplay between inflammation, coagulation, and endothelial damage. Studies have demonstrated that thrombocytopenia correlates with worse outcomes in critically ill patients and serves as an important, albeit non-specific, marker of disease severity [18-20]."

  • To support this statement, we have added the following references, which provide a robust and comprehensive basis for our argument:
  • 18. Costa-Filho, R.C.; Bozza, F.A. Platelets: an outlook from biology through evidence-based achievements in critical care. Ann Transl Med 2017, 5, 449, doi:10.21037/atm.2017.11.04.
  • 19.Levi, M. Platelets in Critical Illness. Semin Thromb Hemost 2016, 42, 252-257, doi:10.1055/s-0035-1570080.
  • 20.Costa-Filho, R. Monitoring the Coagulation. In Controversies in Intensive Care Medicine, Andrew Rhodes, R.K., Marco Ranieri and Rui Moreno, Eds.; Medizinisch Wissenschaftliche Verlagsgesellschaft: Berlin, Germany, 2008; pp. 287–310.

These references emphasize the multifaceted role of platelets in inflammation, coagulation, and critical illness, aligning with the revised discussion.

We sincerely appreciate your thoughtful comments, which have significantly improved the clarity and scientific rigor of our manuscript. Please let us know if you have any further suggestions or concerns.

Thank you again for your time and effort in reviewing our work.

Best regards,
Dr. Rubens Carmo Costa-Filho

Reviewer 2 Report

Comments and Suggestions for Authors

My remarks have been (rapidly!) taken into account thus the revised version is acceptable. Just two observations.

I would have appreciate that the official name of the hMP and the RSV were added.

Just an additional comment": the Astra Zeneca site is poorly informative and more official informations on the vaccine are accessible at the site: https://www.sec.gov/Archives/edgar/data/1786255/000119312523293389/d654575dex992.htm (Icosavax has designed and is the source of the bivalent vaccine).

Author Response

Dear Reviewer,

Thank you very much for your insightful feedback and helpful suggestions. We have carefully addressed your comments in the revised manuscript.

As requested, we have added the official names for human metapneumovirus (hMPV) and respiratory syncytial virus (RSV) in the revised text. Additionally, we incorporated information about the bivalent vaccine IVX-A12, highlighting its Phase II trial results targeting both hMPV and RSV.

We also included a reference to the official site you kindly recommended for more detailed information about the vaccine (Icosavax, Inc.), and it has been cited appropriately in the manuscript ([23]).

The revised section now reads:

"Promising advancements in vaccine development, including virus-like particles, RNA-based platforms, and stabilized fusion proteins, offer hope for effective prevention strategies against human metapneumovirus (hMPV) and related viruses such as respiratory syncytial virus (RSV). For example, investigational bivalent vaccines like IVX-A12, currently in phase II trials and designed to target both hMPV and RSV, highlight ongoing efforts to fill critical gaps in preventing and managing respiratory infections. These efforts underscore the importance of sustained investment in vaccine research to mitigate the global burden of these diseases."

We sincerely appreciate your thorough review and thoughtful recommendations, which have enhanced the clarity and scientific rigor of our manuscript.

Thank you again for your valuable contributions.

Best regards,
Dr. Rubens Carmo Costa-Filho